# High-Yield Production of PCV2 Cap Protein: Baculovirus Vector Construction and Cultivation Process Optimization

**DOI:** 10.3390/vaccines13080801

**Published:** 2025-07-28

**Authors:** Long Cheng, Denglong Xie, Wei Ji, Xiaohong Ye, Fangheng Yu, Xiaohui Yang, Nan Gao, Yan Zhang, Shu Zhu, Yongqi Zhou

**Affiliations:** 1Department of Veterinary Medicine, College of Animal Sciences, Zhejiang University, Hangzhou 310058, China; chenglongljt@163.com; 2Zhejiang Hisun Animal Healthcare Products Co., Ltd., Hangzhou 311400, China; denglong.xie@hisunah.com (D.X.); wei.ji@hisunah.com (W.J.); xiaohong.ye@hisunah.com (X.Y.); fangheng.yu@hisunah.com (F.Y.); xiaohui.yang01@hisunah.com (X.Y.); nan.gao@hisunah.com (N.G.); yan.zhang@hisunah.com (Y.Z.)

**Keywords:** porcine circovirus type 2 (PCV2), baculovirus, insect expression system, *flash*BAC, Bac-to-Bac

## Abstract

**Background/Objectives:** Porcine circovirus type 2 (PCV2) infection causes porcine circovirus disease (PCVD), a global immunosuppressive disease in pigs. Its clinical manifestations include post-weaning multisystemic wasting syndrome (PMWS) and porcine dermatitis and nephropathy syndrome (PDNS), which cause significant economic losses to the swine industry. The Cap protein, which is the major protective antigen of PCV2, can self-assemble to form virus-like particles (VLPs) in the insect baculovirus expression system. Few studies have compared the expression of Cap proteins in different baculovirus expression systems. **Methods:** In this study, we compared two commonly commercialized baculovirus construction systems with the Cap protein expression in various insect cells. **Results:** The results demonstrate that the *flash*BAC system expressed the Cap protein at higher levels than the Bac-to-Bac system. Notably, when expressing four copies of the Cap protein, the *flash*BAC system achieved the highest protein yield in High Five cells, where it reached 432 μg/mL at 5 days post-infection (dpi) with 27 °C cultivation. Animal experiments confirmed that the purified Cap protein effectively induced specific antibody production in mice and swine. **Conclusions:** This study provides critical data for optimizing the production of the PCV2 Cap protein, which is of great significance for reducing the production cost of PCV2 vaccines and improving the industrial production efficiency.

## 1. Introduction

Circoviruses are recognized as the smallest viruses known to infect animals. Four types of porcine circoviruses (PCV1–PCV4) have been sequentially identified in swine [1]. Porcine circovirus type 1 (PCV1), initially identified as the first member of the porcine circovirus family, has been characterized as a nonpathogenic virus [2]. PCV2 remains the main pathogen of related diseases. The pathogenicity of PCV3 remains controversial. PCV4 is a recently discovered pathogen, and its pathogenicity in pigs requires further investigation [3,4]. PCV2 infection causes porcine circovirus disease (PCVD), a global immunosuppressive disease in pigs. The clinical manifestations of this disease include post-weaning multisystemic wasting syndrome (PMWS) and porcine dermatitis and nephropathy syndrome (PDNS), which cause significant economic losses to the swine industry [5,6]. PCV2 is classified into eight genotypes (PCV2a–PCV2h) [7,8]. Recently, a novel genotype, PCV2i, was identified [9]. PCV2d is the predominant circulating subtype of PCV2 [5]. The PCV2 genome is a covalently closed circular single-stranded DNA (ssDNA) that has approximately 1.7 kilobases (kb) [10]. Software analysis has identified 11 open reading frames (ORFs) in the PCV2 genome, but only 4 are utilized for protein expression [11]. ORF2 encodes the sole structural protein—the capsid (Cap) protein. The Cap protein represents the primary antigenic determinant of the virus, which serves as a critical determinant of the host immune response and vaccine efficacy [12].

For the prevention of PCVD, vaccination remains the primary strategy [6]. Virus-like particles (VLPs), as a type of subunit vaccine, have emerged as the preferred candidate due to their lack of nucleic acids, inability to replicate autonomously, and enhanced safety profile. The Cap protein can self-assemble into VLPs across various expression systems [13]. Although bacterial, yeast, and insect cell platforms can express the Cap protein, commercially available vaccines (e.g., Ingelvac CircoFLEX^®^ and Circumvent^®^) exclusively utilize the baculovirus–insect cell system [14,15,16]. This preference is likely attributed to its advantages in high-yield production, appropriate post-translational modifications, and efficient VLP assembly. While alternative expression systems, such as *E. coli* and yeast, are being explored for their cost-effectiveness, they currently face challenges related to immunogenicity and scalability limitations [17,18].

The Bac-to-Bac and *flash*BAC systems are two commonly commercialized baculovirus construction systems. Each system has its own advantages and limitations in terms of the operation procedures and application scope [19,20]. The *flash*BAC system offers several modified baculoviral backbone vectors. The original *flash*BAC vector has the chiA (chitinase) gene deleted. The *flash*BAC Gold vector removes both the chiA and v-cath (cathepsin) genes. The *flash*BAC Ultra vector further deletes additional viral genes (such as p10, p74, and p26) beyond chiA and v-cath. These modifications reduce recombinant protein degradation and enable prolonged protein expression. In the Bac-to-Bac system, the target gene is first cloned into a pFast series vector. The recombinant plasmid is then transformed into *E. coli* DH10Bac-competent cells to generate the bacmid through site-specific transposition. Positive clones are selected via blue–white screening, followed by bacmid extraction and transfection into insect cells for viral packaging (Figure 1A) [21]. In contrast, the *flash*BAC system employs compatible transfer vectors (e.g., pBAC or pOET series) to construct the expression cassette. The recombinant transfer vector is co-transfected with a linearized bacmid into insect cells, where homologous recombination occurs in vivo to directly generate the recombinant baculovirus (Figure 1B) [19,22]. Although both systems are commercial, laboratories choose one system based on their existing platform and research costs and seldom compare the two systems side by side [20]. Although numerous studies have documented the application of both systems for Cap protein expression, applications of both systems with simultaneous Cap protein expression comparisons remain unexplored. For certain proteins, increasing the gene copy number can enhance the expression level of recombinant proteins [23,24]. It remains unknown whether increasing the copy number of the *cap* gene enhances the protein expression in the insect baculovirus system.

In this study, we compared two commonly commercialized baculovirus construction systems for expressing the Cap protein in various insect cells. Our findings demonstrate that the *flash*BAC system achieved significantly higher Cap protein expression levels compared with the Bac-to-Bac system, where an increased gene copy number of the Cap protein further enhanced the expression efficiency. The High Five insect cells demonstrated a superior performance for Cap protein expression compared with the Sf9 cells. Notably, when expressing four Cap protein copies, the *flash*BAC system achieved the highest protein yield of 432 μg/mL in the High Five cells. Animal immunization experiments confirmed that the expressed Cap protein effectively induced specific antibody production in both mice and swine. This study provides critical data for optimizing PCV2 Cap protein production, which is of great significance for cost reduction in porcine circovirus vaccine manufacturing.

## 2. Materials and Methods

### 2.1. Vector Construction

The target gene Cap (GenBank accession no. MK751862.1) was codon-optimized for expression in *Spodoptera frugiperda* cells and synthesized by Zhejiang Sunya Biotechnology Co., Ltd. (Zhejiang, China). The optimized DNA sequence was detailed in Additional file 1: Appendix A. Two expression systems were employed: the pFast-Dual vector (Bac-to-Bac system) and the pBAC-1 vector (*flash*BAC system). pFast-Cap vector construction: The synthesized Cap gene was used as a template for polymerase chain reaction (PCR) amplification using PrimeSTAR^®^ Max DNA Polymerase (Takara, R045Q, Shiga, Japan) with specific primers (pFast-Cap-F/R). The target fragment was purified using 1% agarose gel electrophoresis. The pFast-Dual vector was double-digested with *Bam*HI (Takara, 1605, Shiga, Japan) and *Hin*dIII (Takara, 1615, Shiga, Japan), and the linearized vector was gel-purified. The insert and vector were ligated at a 3:1 molar ratio (insert/vector) using the ClonExpress Ultra One-Step Cloning Kit (Vazyme, C115-01, Nanjing, China) and transformed into DH5α-competent cells (TransGen Biotech, CD201, Beijing, China). Positive clones were selected on Luria–Bertani (LB) agar plates that contained 100 μg/mL ampicillin (Sangon Biotech, A610028, Shanghai, China) after an overnight incubation at 37 °C. Colony PCR and sequencing were performed for validation, and the positive recombinant plasmid was extracted using a Plasmid Extraction Kit (Tiangen, DP106, Beijing, China). pBAC-1-Cap vector construction: The same strategy was applied, where the Cap gene was amplified using pBAC-1-Cap-F/R primers, and the pBAC-1 vector was digested with *Bam*HI/*Hin*dIII for subsequent cloning. pBAC-4×Cap vector construction: A pUC19-4×Cap recombinant vector was synthesized by Zhejiang Sunya Biotechnology Co., Ltd. (Zhejiang, China). The multi-copy Cap gene fragment was obtained using double digestion with *Bgl*II (Takara, 1606, Shiga, Japan) and *Hin*dIII and then ligated into the similarly digested pBAC-1 vector using T4 DNA Ligase (NEB, M0202V, Ipswich, MA, USA) at a 5:1 insert-to-vector ratio. The ligation product was transformed into Sure competent cells (WEIDI, DL1065S, Shanghai, China). Positive clones were verified via sequencing, and the plasmid was extracted for downstream applications. All primer sequences used for vector construction were detailed in Additional file 2: Appendix A.

### 2.2. Bacmid Construction

The Bac-to-Bac method requires the construction of bacmid via transfecting DH10Bac chemically competent cells (Weidi, DL1071, Shanghai, China) with an expression plasmid. The pFast-Cap plasmid was transformed into DH10Bac chemically competent cells, and the heat-shocked cells were plated on Luria–Bertani (LB) agar plates that contained 50 μg/mL kanamycin (Sangon Biotech, A506636, Shanghai, China), 7 μg/mL gentamicin (Sangon Biotech, A428430, Shanghai, China), 7 μg/mL tetracycline (Sangon Biotech, A430165, Shanghai, China), 40 μg/mL X-gal (Sangon Biotech, A600083, Shanghai, China), and 40 μg/mL IPTG (Sangon Biotech, A600168, Shanghai, China). Positive white colonies were selected via blue–white screening and further verified via PCR using pFast-Cap-F/R, M13(40)/Tn7R, and Tn7L/M13R primers [21]. The bacmid DNA was extracted using the BAC/PAC DNA Kit (Omega Bio-Tek, D2156, Norcross, Georgia, USA) for subsequent baculovirus packaging. All primer sequences used for bacmid construction were detailed in Additional file 2: Appendix A.

### 2.3. Baculovirus Packaging and Amplification

The Sf9 cells (cell lines were provided by Suzhou Womei Biology Co., Ltd. (Suzhou, China)) were cultured in an SF-SMF cell culture medium in a 27 °C incubator. We ensured the cells were in the logarithmic growth phase (density of approximately 1–2 × 10^6^ cells/mL). In total, 2.5 × 10^6^ cells were seeded in a 6-well plate with a culture volume of 2 mL. After 1 h, the medium was replaced with 2.5 mL of Transfection Media (Oxford Expression Technologies, 500312, Oxford OX3 0BP, UK). For the Bac-to-Bac system, 8 μg of Bacmid was added to 100 μL of Transfection Media, followed by 1.2 μL of BaculoFECTIN II (Oxford Expression Technologies, 300105, Oxford OX3 0BP, UK). For the *flash*BAC system, 100 ng of *flash*BAC GOLD baculovirus DNA (Oxford Expression Technologies, 100202, Oxford OX3 0BP, UK) and 500 ng of pBAC-1-Cap/pBAC4×-1-Cap was added to 100 μL of Transfection Media, followed by 1.2 μL of BaculoFECTIN II (Oxford Expression Technologies, 300105, Oxford OX3 0BP, UK). The mixture was incubated at room temperature for 15 min before being added to the 6-well plate. After 5 days, the cell supernatant was collected to obtain the P1 virus generation and stored at −80 °C. To obtain the P2 virus generation, the Sf9 cells were inoculated at a density of 2 × 10^6^ cells/mL with a multiplicity of infection (MOI) = 0.05 of the P1 baculovirus stock. The cell supernatant was collected when the cell viability reached 50%. The P3 virus generation was obtained using the same method as for the P2 generation.

### 2.4. Baculovirus Identification and Titer Test

The second-generation (P2) baculovirus DNA was extracted using a viral genome DNA/RNA extraction kit (Tiangen, DP315, Beijing, China). PCR amplification was performed using baculovirus VP80-specific primers, and the target band was detected using agarose gel electrophoresis. The viral titer was determined using a GP64-specific antibody against the baculovirus. The Sf9 cells were seeded in a 96-well plate at a density of 7 × 10^6^ cells diluted in 10 mL of medium, with 100 μL of the cell suspension added to each well. After incubation at 27 °C for 30 min, the viral stock was serially diluted in 10-fold increments across 10 gradients. Each dilution was added to a column of 8 wells in the 96-well plate, with the last two columns left as blanks. The plate was incubated at 27 °C for 4 days. Subsequently, the cells were fixed with 4% paraformaldehyde, permeabilized with 0.3% Triton X-100 in phosphate-buffered saline (PBS), and blocked with 3% BSA in PBS. The cells were then incubated with a 1:1000 dilution of GP64 antibody (Santa Cruz, sc-65499, Dallas, TX, USA) at 37 °C for 2 h, followed by three washes. A 1:2000 dilution of anti-mouse IgG (H + L) cross-adsorbed secondary antibody conjugated with Alexa Fluor™ 488 (Thermo, A-11001, Waltham, MA, USA) was added, and the mixture was incubated at 37 °C for 2 h, followed by another three washes. The positive wells were observed under a fluorescence microscope, and the viral titer (TCID_50_/mL) was calculated using the Reed and Muench method.

### 2.5. The Cap Protein Expression Was Detected Using SDS-PAGE and ELISA

Sf9 cells and High Five cells (cell lines were provided by Suzhou Womei Biology Co., Ltd. (Suzhou, China)) were cultured in SF-SMF medium at 27 °C. Cells in the logarithmic growth phase were used for infections. A total of 2.0 × 10^6^ cells were seeded into a 125 mL flask containing 25 mL of culture medium and infected at a multiplicity of infection (MOI) of 0.05 using a P3 baculovirus stock. Cultures were maintained at 27 °C with shaking at 110 rpm. Cell growth parameters were monitored daily, and samples were collected for analysis by SDS-PAGE and ELISA. The protein sample was treated with protein loading buffer (TransGen Biotech, DL101-02, Beijing, China) at 95 °C for 5 min. The treated sample was then loaded onto a 12% SDS-PAGE gel (Yamei, PG113, Shanghai, China), and electrophoresis was performed at 80 V for 30 min, followed by 120 V for 1 h and 20 min. The protein bands were visualized by staining with Coomassie Brilliant Blue R-250. The concentration of the protein sample was measured using the Quantitative Determination for the Total Concentration of the PCV2-Cap Protein ELISA Kit (Womei, FGPC292002, Suzhou, Jiangsu, China), with the specific methodology detailed in the manufacturer’s instructions.

### 2.6. Cap Protein Purification and Identification

Purification of the Cap protein based on its properties primarily employs cation exchange chromatography. A chromatography column was packed with SP Sepharose Fast Flow cation exchange resin (Bestchrom, AI0011, Jiaxing, Zhejiang, China) and equilibrated with 5 column volumes (CV) of equilibration buffer (20 mM sodium phosphate, pH 6.0) until the effluent’s pH and conductivity stabilized. The insect cell culture supernatant was centrifuged (12,000× *g*, 30 min, 4 °C), and the supernatant was filtered through a 0.45 μm membrane (Millipore, SLHP033N, Burlington, MA, USA). The sample was then diluted with an equilibration buffer to achieve a conductivity below 5 mS/cm, and the pH was adjusted to 6.0. The processed sample was loaded onto the equilibrated cation exchange column at a 1 mL/min flow rate to ensure efficient binding of the target protein. The column was washed with 10 CV of equilibration buffer to remove unbound impurities until the UV absorbance (280 nm) returned to the baseline. A linear gradient elution was performed using an equilibration buffer that contained 1 M NaCl (Sangon Biotech, A610476, Shanghai, China) at a 1 mL/min flow rate. The eluted peaks were collected while monitoring the UV absorbance (280 nm). The fractions that corresponded to the elution peak were pooled and analyzed for target protein purity using SDS-PAGE and a Western blot. For the Western blotting, proteins from an unstained PAGE gel were transferred to a PVDF membrane (Biorad, 1620177, Hercules, CA, USA) using a wet transfer system. The membrane was blocked with 5% skim milk (Oxoid, LP0031B, Waltham, MA, USA) in PBS for 2 h, incubated overnight at 4 °C with a 1:1000 dilution of primary antibody against PCV2 (Bioss, bs-10057R, Beijing, China) in PBS, washed five times with PBS-Tween, incubated with a 1:5000 dilution of anti-rabbit secondary antibody (Thermo, 31460, Waltham, MA, USA) at 37 °C for 2 h, washed five times with PBS-Tween, and finally developed with an ECL substrate (Vazyme, E423-01/02, Shanghai, China) before being imaged with a gel imaging system.

### 2.7. Animal Immunity and Efficacy Evaluation

The immunogenicity of the Cap protein was evaluated in mice and pigs. Six-week-old BALB/c mice (Sipeifu, Beijing, China) were randomly divided into two groups (n = 5 per group). The vaccine group was immunized with 50 μg of Cap protein mixed with the oil-in-water emulsion adjuvant SDA 15A (Merchinade, SDA15AVG, Beijing, China) at a 4:1 ratio (protein:adjuvant, *v*/*v*). The control group received phosphate-buffered saline (PBS) mixed with the same adjuvant at a 4:1 ratio (PBS:adjuvant, *v*/*v*), administered at the same volume as the vaccine group. The immunization method for mice was subcutaneous injection. Blood was collected from the mice 5 weeks post-immunization, and the antibody titers in the sera were determined. Healthy weaned piglets aged 2 weeks were randomly divided into two groups, with 10 piglets in each group. The vaccine group was immunized with 100 μg of Cap protein mixed SDA 15A adjuvant at a ratio of 4:1 (protein:adjuvant, *v*/*v*), while the control group was immunized with the same volume PBS mixed with the same adjuvant at a 4:1 ratio (PBS/adjuvant, *v*/*v*). The immunization method for pigs was intramuscular injection. Serum samples were collected at week 0 (pre-immunization) and weeks 4, 7, 10, 13, 16, and 19 (post-immunization), and the antibody titers were measured. The antibody titers in the piglet serum were determined by indirect ELISA (INGENASA, 11.PCV.K.1/5, Madrid, Spain) to detect the Cap-protein-specific antibodies. A sample was considered positive if the OD 450 nm value was ≥0.5 and at least 2.1 times higher than the negative control. The antibody titers in the mouse sera were measured using indirect ELISA (INGENASA, 11.PCV.K.1/5, Madrid, Spain), with the secondary antibody replaced by an HRP-labeled anti-mouse secondary antibody (Thermo, G-21040, Waltham, MA, USA).

## 3. Results

### 3.1. Expression Vector and Bacmid Construction

This experiment required the construction of three vectors: pFast-Cap, pBAC-1-Cap, and pBAC-4×Cap. Among them, pFast-Cap needed to be transfected into DH10Bac to generate the bacmid, where the construction schematic is shown in Figure 2A. pFast-Cap vector construction: The synthesized Cap gene was used as a template, and PCR amplification was performed with specific primers (pFast-Cap-F/R). The target fragment size was 746 bp, as confirmed using 1% agarose gel electrophoresis (Figure 2B). The pFast-Dual vector was double-digested with *Bam*HI/*Hin*dIII, which yielded a target fragment of 5166 bp (Figure 2C). pBAC-1-Cap vector construction: Using the synthesized Cap gene as a template, the target gene was amplified with the pBAC-1-Cap-F/R primers, which resulted in a 749 bp product (Figure 2D). The pBAC-1 vector was double-digested with *Bam*HI/*Hin*dIII, which produced a 5234 bp target fragment (Figure 2E). pBAC-4×Cap vector construction: The recombinant vector was synthesized by Shangya Biotechnology. The multi-copy Cap gene fragment (3753 bp, Figure 2F) was obtained using the *Bgl*II/*Hin*dIII double-digestion of pUC19-4×Cap and ligated into the similarly digested pBAC-1 vector (5132 bp, Figure 2G). The pFast-Cap plasmid was transformed into DH10Bac-competent cells, which yielded distinct blue and white colonies (Figure 2H). The white colonies were positive clones, and PCR screening was performed using three primer pairs: M13 (40)/TnR-R, Cap-F/R, and TnL-F/M13R, which successfully amplified the target bands (Figure 2I).

### 3.2. Baculovirus Identification and Titration

To determine whether the baculovirus was successfully packaged, the DNA of the packaged viral cells was extracted and detected using VP80-specific primers. The PCR results show that baculovirus-specific bands, with a size of 193 bp, were successfully detected using the baculovirus VP80-specific primers (Figure 3A), indicating that the baculovirus was successfully packaged. The 50% tissue culture infectious dose (TCID_50_) of the baculovirus was measured via immunofluorescence staining using a baculovirus GP64-specific antibody. The results show that the viral titers of pFast-Cap, pBAC-1, and pBAC4×-1 were 7.35 log_10_TCID_50_/mL, 8.41 log_10_TCID_50_/mL, and 8.01 log_10_TCID_50_/mL, respectively (Figure 3B,C).

### 3.3. Growth Parameters of Sf9 and High Five Cells After Baculovirus Infection

To determine the optimal time for harvesting the protein, the viable cell counts, total cell counts, viabilities, and cell diameters of the Sf9 and High Five insect cells infected with baculovirus were measured every 24 h until the cell viability dropped below 20%. The total Sf9 cell count initially increased and then decreased. On the first day post-infection, the cells proliferated normally, but after 24 h, the proliferation rate of the infected group was significantly lower than that of the blank control group (Figure 4A). The Sf9 viable cell count also showed an initial increase during days 1–2 followed by a decrease after day 3 (Figure 4B). The Sf9 cell viability remained above 90% for the first three days but dropped by the fourth day. By the fifth day, the viability of some infected groups fell to 20%, whereas the control group also exhibited a decline but at a significantly slower rate compared with the infected group (Figure 4C). From days 1 to 3 post-infection, the Sf9 cell diameter gradually increased, with the infected group showing a faster rate of increase than the control group; however, the diameter decreased on days 4–5 (Figure 4D). The High Five total cell count increased with the infection time and peaked on the fifth day, though the proliferation rate of the infected group was notably lower than that of the control group (Figure 4E). The High Five viable cell count exhibited an initial rise during days 1–2, followed by a decline after day 3, similar to the trend observed in the Sf9 cells (Figure 4F). The High Five cell viability remained above 90% for the first three days but decreased by the fourth day, where some infected groups dropped to 20% viability by the fifth day (Figure 4G). On day 1 post-infection, the High Five cell diameter decreased, but from days 2 to 3, it gradually increased, where the infected group displayed a faster rate of enlargement than the control. By days 4–5, the cell diameter had decreased again (Figure 4H).

### 3.4. Analysis of Cap Protein Expression in Baculovirus-Infected Sf9 and High Five Cell Lines

The constructed baculovirus pFastBac-Cap infected Sf9 cells and the expression level of Cap protein increased with the number of infection days, which reached 164 μg/mL at 5 dpi. The expression of the Cap protein in the pFastBac-Cap-infected High Five cells also increased with the infection time, which peaked at 129 μg/mL at 5 dpi (Figure 5A,B). For the baculovirus pBAC-1-Cap-infected Sf9 cells, the Cap protein expression rose with prolonged infection, which reached its highest level of 298 μg/mL at 4 dpi. Similarly, in pBAC-1-Cap-infected High Five cells, the Cap protein expression increased over time, with the highest level of 392 μg/mL observed at 5 dpi. In the case of pBAC-4×Cap baculovirus-infected Sf9 cells, the Cap protein expression initially increased and then decreased, which peaked at 319 μg/mL at 4 dpi before it declined to 287 μg/mL at 5 dpi (Figure 5A,B). For the pBAC-4×Cap-infected High Five cells, the Cap protein expression showed a similar trend, where it rose initially and then decreased, with the highest expression level of 431 μg/mL at 4 dpi, followed by a drop to 395 μg/mL at 5 dpi (Figure 5A,B).

### 3.5. Cap Protein Purification

To improve the purity of Cap, cell debris was removed using centrifugation, the culture supernatant was clarified using a 0.45 μm filter, and the Cap protein was finally purified by cation-exchange chromatography. The purified Cap was detected via Western blotting using a specific antibody. We found that centrifugation effectively removed the cell debris, and the cation-exchange purification yielded a higher concentration of Cap protein with a one-step elution that showed better results (Figure 6A). The Western blot analysis of the purified protein using a Cap-specific antibody confirmed specific binding (Figure 6B). The results demonstrate that the cation-exchange chromatography could successfully purify the protein, and the purified product was the Cap protein.

### 3.6. Evaluation of Immune Effect of Cap Protein

The purified Cap protein was emulsified with adjuvant and used to immunize mice and pigs. Serum samples were collected at designated time points, and anti-Cap antibody titers were measured using ELISA (Figure 7A). In the mice, Cap-specific antibodies were detectable at 5 weeks post-immunization, with a peak titer of 1:12,800 (Figure 7B). In the pigs, the antibody titers increased progressively from weeks 4 to 7 but declined thereafter, where they reached levels comparable with the control group by week 10 and remained undetectable through to week 19 (Figure 7C). These findings demonstrate that the Cap protein effectively induced an immune response that elicited robust but transient antibody production in immunized animals.

## 4. Discussion

PCV2 infection causes PCVD, a global immunosuppressive disease in pigs. Its clinical manifestations include PMWS and PDNS, which cause significant economic losses to the swine industry [5,6]. Vaccination remains the primary strategy for preventing PCVD [6]. Cap, as the protective antigen of porcine circovirus, has been expressed in various protein expression systems, such as *E. coli*, yeast, and the insect baculovirus system [25,26,27]. However, currently marketed products are exclusively expressed using the insect baculovirus system, indicating its superior suitability for protein expression. Common baculovirus construction systems include Bac-to-Bac and the homologous recombination method *flash*BAC [19,20], yet few laboratories have compared the impact of different baculovirus construction systems on protein expression. The *flash*BAC system offers several modified baculoviral backbone vectors. The original *flash*BAC vector has the chiA (chitinase) gene deleted. The *flash*BAC Gold vector removes both the chiA and v-cath (cathepsin) genes. The *flash*BAC Ultra vector further deletes additional viral genes (such as p10, p74, and p26) beyond chiA and v-cath. These modifications reduce recombinant protein degradation and enable prolonged protein expression. Our results demonstrate that the baculovirus constructed using *flash*BAC was more suitable for Cap protein expression than that constructed using the Bac-to-Bac system, regardless of whether Sf9 or High Five cells were employed. The cause of this result may be related to the use of the *flash*BAC Gold vector in this experiment. Research has shown that the presence of the chiA and v-cath genes can promote cell lysis, leading to a shortened protein expression time and a significant decrease in protein expression levels [28,29]. Furthermore, the insect High Five cell line was more conducive to Cap protein expression compared with Sf9 cells. The literature and patents report protein expression levels ranging from 50 to 150 μg/mL, with the highest patent-reported level reaching 200 μg/mL [30]. In this study, the maximum expression level achieved was 432 μg/mL, significantly surpassing the values reported in the literature and patents.

In addition to codon optimization, increasing the gene copy number can enhance protein expression [23,31]. In this study, the *flash*BAC system was used to construct baculoviruses that carried one-copy and four-copy Cap protein genes. We found that elevating the copy number could improve the Cap protein expression level, but not to the extent of a fourfold increase—only a 12% enhancement compared with the one-copy construct was achieved. This limited improvement may have been due to the restricted expression capacity of insect cells or the presence of negative feedback mechanisms. Meanwhile, the baculovirus that expressed four copies of the Cap protein reached its peak expression on the fourth day post-infection, followed by a decline on the fifth day. This reduction might be attributed to cellular negative feedback mechanisms or the onset of cell lysis, releasing intracellular proteases that degrade the protein. Although the Cap protein in this experiment was designed for intracellular expression and its levels decreased by the fifth day, harvesting at this time point was deemed more suitable for industrial-scale downstream purification. This is because a higher proportion of cell death at this stage facilitates the subsequent purification process.

In addition to codon optimization, increasing the gene copy number can enhance protein expression [32]. In this study, the *flash*BAC system was used to construct baculoviruses carrying one-copy and four-copy genes of the Cap protein. We found that elevating the copy number could improve the expression level of the Cap protein, but not to the extent of a fourfold increase—only a 12% enhancement compared to the one-copy construct. This limited improvement might be attributed to the constrained expression capacity of insect cells or the presence of negative feedback mechanisms [33]. Furthermore, the baculovirus with four copies of the Cap protein reached peak expression on the 4 dpi, followed by a decline on the fifth day. This reduction could be due to cellular negative feedback mechanisms or the onset of cell lysis, releasing intracellular proteases that degrade the protein. Although the Cap protein in this study was designed for intracellular expression and its levels decreased by the fifth day, harvesting at this time point was deemed more suitable for industrial-scale downstream purification. This is because a higher proportion of cell death at this stage facilitates the subsequent purification process.

Baculoviruses also exhibit passage instability. With successive viral passages, an increasing number of defective viruses that are incapable of expressing the target protein are generated. These non-functional viral particles proliferate at a higher rate compared with viruses carrying the target gene. Consequently, the expression levels of the target protein progressively declines with extended passaging [34]. Both industrial production guidelines and the scientific literature therefore recommend limiting baculovirus passages to a maximum of five generations to maintain optimal protein expression. While increasing the copy number can enhance the expression, it also raises the risk of homologous recombination in the viral genome. Thus, the probability of viral genome recombination increases with more passages [35]. To ensure stable protein expression, it is advisable to restrict baculovirus passages to 4–5 generations. This limited number of passages effectively mitigates the risk of viral genome instability caused by increased copy numbers.

MOI optimization in baculovirus infection is also an important factor in improving protein expression [36]. In this experiment, the MOI was not optimized, and the recommended MOI of 0.05 from the kit was used. Future studies will consider MOI optimization to further enhance the Cap protein expression level. In addition to enhancing the protein yield by optimizing the codon usage of the target protein to match that of insect cells [29], promoter engineering is another frequently employed strategy to improve protein expression. Studies have shown that cis-linking the polyhedrin promoter to the p10 promoter enhances protein production, while co-expressing molecular chaperones and transcription-associated proteins, such as baculovirus transactivation factors IE1 and IE0, can improve baculovirus protein expression levels [37]. This study did not explore the optimization of promoter activity or the expression of molecular chaperones. Whether the use of tandem promoter combinations or the co-expression of molecular chaperones could further enhance protein expression warrants further investigation.

This study demonstrated the superior performance of the *flash*Bac system over the Bac-to-Bac system in the expression efficiency of PCV2 Cap protein. It should be noted that the *flash*Bac Gold vector used in this experiment had the chiA and v-cath genes deleted, whereas these genes were retained in the Bac-to-Bac system. Previous studies have shown that the deletion of chiA and v-cath can significantly improve the yield and stability of recombinant proteins, which may partially account for the observed advantage of the *flash*Bac system. To more accurately evaluate the inherent performance differences between baculovirus packaging systems, follow-up studies should use MultiBac or EMBacY systems (which share similar construction methods with Bac-to-Bac), but with comparable deletions of chiA and v-cath genes, for parallel comparison with the *flash*Bac system [20]. Through such rigorously controlled experimental designs, we can distinguish whether the observed advantages stem directly from gene deletion effects or truly reflect the efficiency differences of the viral packaging systems themselves. Ultimately, these follow-up investigations will provide more reliable evidence for researchers to select the optimal expression platform.

## 5. Conclusions

This study utilized different baculovirus construction systems, Bac-to-Bac and *flash*BAC, to generate recombinant baculoviruses that expressed the Cap protein. Additionally, the *flash*BAC system was employed to construct baculoviruses with varying Cap protein copy numbers. The Cap protein expression levels were evaluated by infecting different insect cell lines: High Five and Sf9. The results demonstrate that the recombinant baculovirus constructed using the *flash*BAC system yielded the highest Cap protein expression when infecting High Five cells. Furthermore, our findings reveal that increasing the *cap* gene copy number in the baculovirus enhanced the Cap protein expression levels. This study provides critical data for optimizing the Cap protein production process, which holds significant importance for reducing the production cost of porcine circovirus vaccines and improving vaccine manufacturing efficiency.

## Figures and Tables

**Figure 1 vaccines-13-00801-f001:**
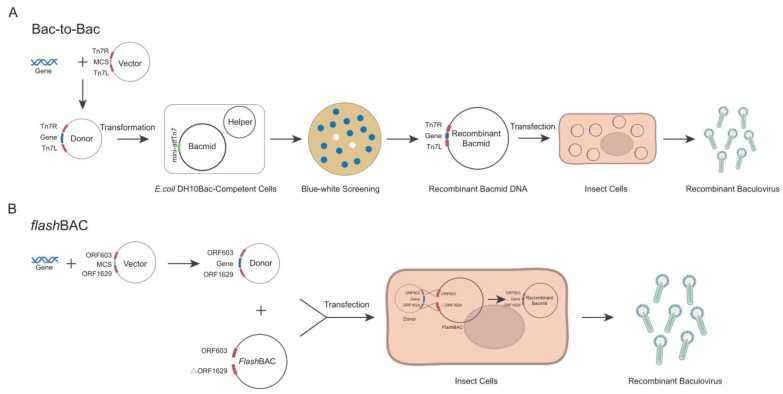
Schematic diagram of Bac-to-Bac and *flash*BAC construction of baculovirus.

**Figure 2 vaccines-13-00801-f002:**
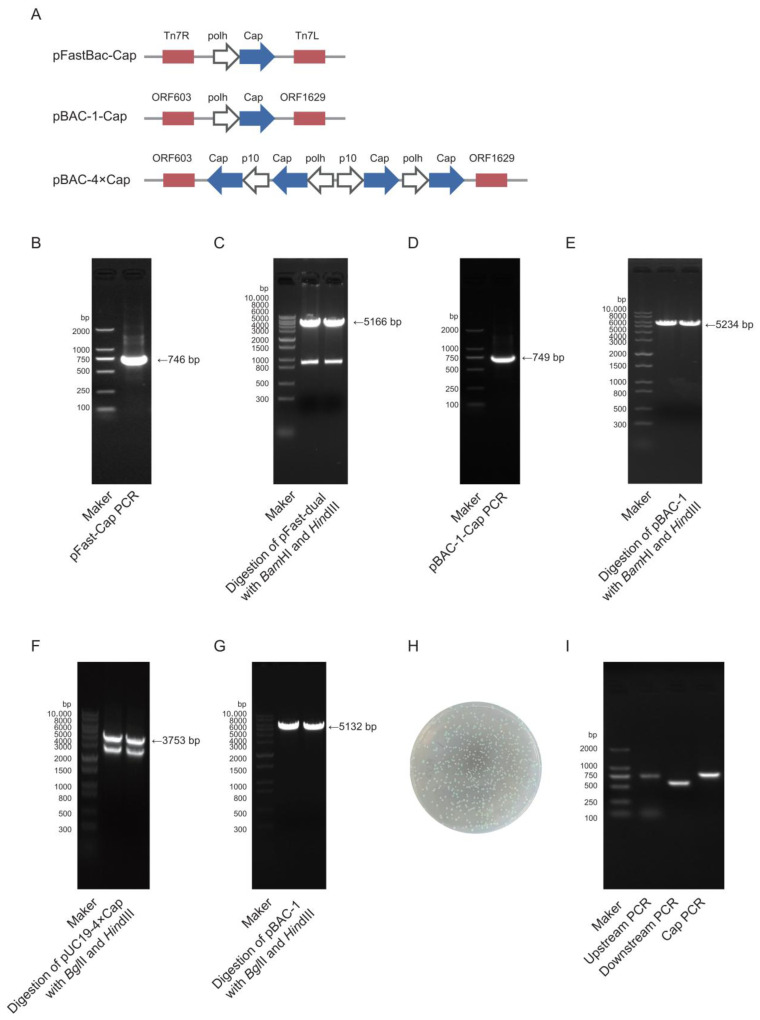
Construction of pFast-Cap, pBAC-1-Cap, and pBAC-4×Cap vectors. (**A**) Gene expression cassette schematic. (**B**) pFast-Cap PCR. (**C**) pFast-dual digestion with *Bam*HI and *Hin*dIII. (**D**) pBAC-Cap PCR. (**E**) pBAC-1 digestion with *Bam*HI and *Hin*dIII. (**F**) pUC19-4×Cap digestion with *Bgl*II and *Hin*dIII. (**G**) pBAC-1 digestion with *Bgl*II and *Hin*dIII. (**H**) Transformation of DH10Bac *E. coli*-competent cells with pFast-Cap. (**I**) PCR-based identification of bacmid.

**Figure 3 vaccines-13-00801-f003:**
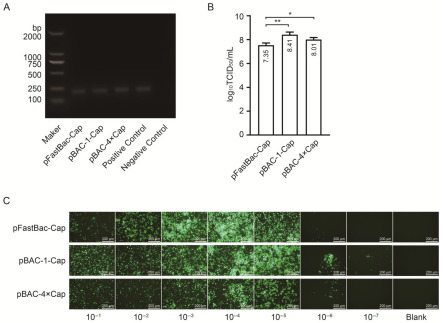
Baculovirus identification via PCR and titer determination using immunofluorescent staining. (**A**) Baculovirus identification using VP80 primers. (**B**) Baculovirus TCID_50_. (**C**) Baculovirus immunofluorescence staining. Asterisks indicate statistically significant differences (* *p* < 0.05, ** *p* < 0.01) by one-way ANOVA.

**Figure 4 vaccines-13-00801-f004:**
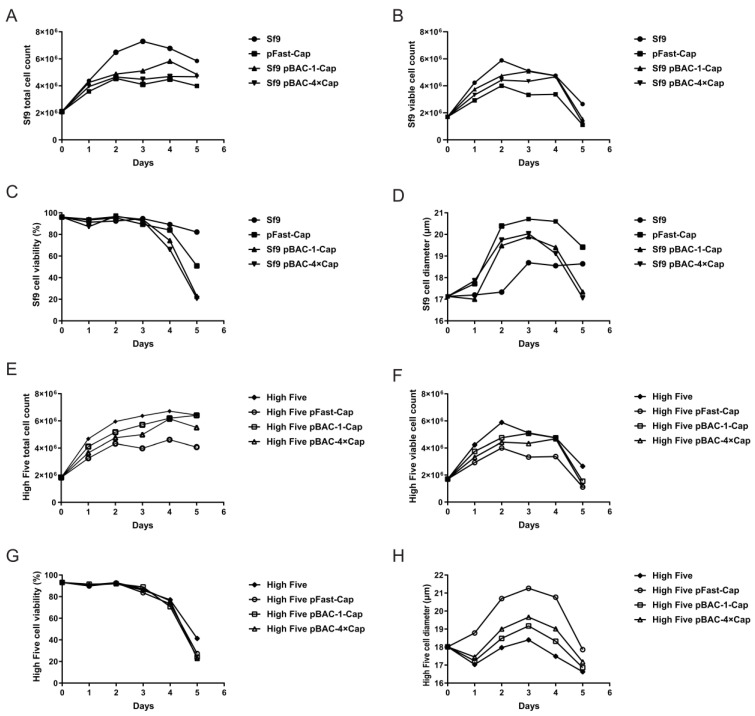
Growth parameters of Sf9 and High Five cells after baculovirus infection. (**A**) Total Sf9 cell number post-infection. (**B**) Viable Sf9 cell number post-infection. (**C**) Sf9 cell viability post-infection. (**D**) Sf9 cell diameter post-infection. (**E**) Total High Five cell number post-infection. (**F**) Viable High Five cell number post-infection. (**G**) High Five cell viability post-infection. (**H**) High Five cell diameter post-infection.

**Figure 5 vaccines-13-00801-f005:**
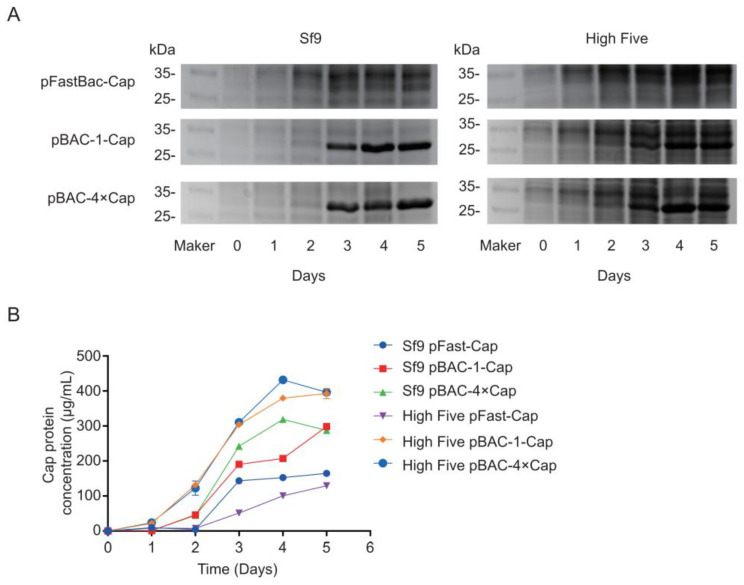
Cap protein expression in baculovirus-infected Sf9 and High Five cell lines. (**A**) Expressed Cap protein analysis using SDS-PAGE. (**B**) Cap protein concentration determination using ELISA.

**Figure 6 vaccines-13-00801-f006:**
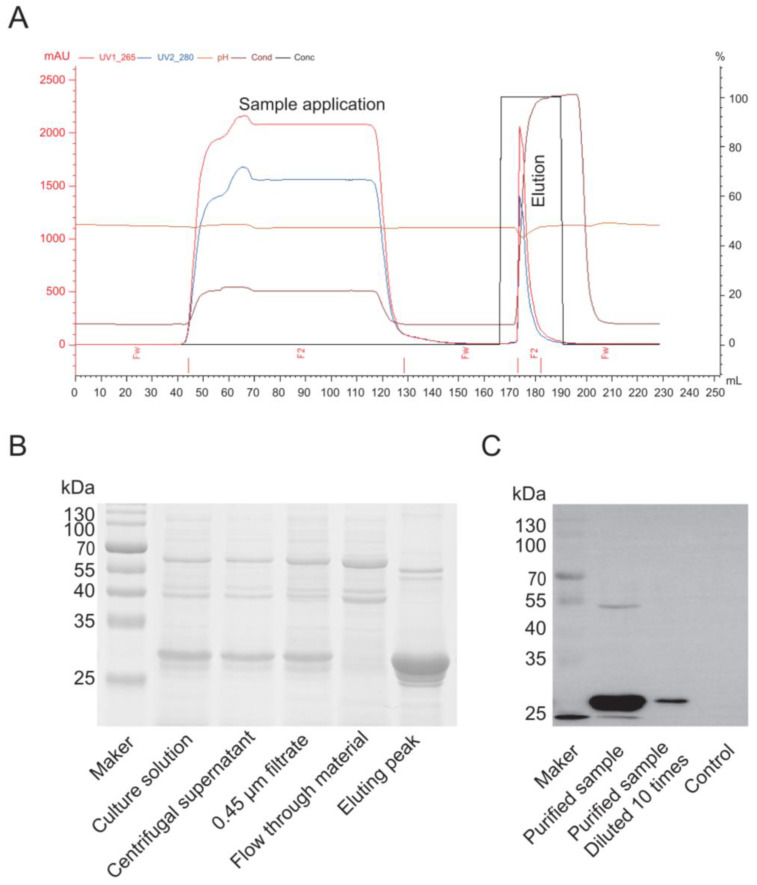
Cap protein purification using cation-exchange chromatography. (**A**) Cation exchange chromatography profile. (**B**) Cap protein analysis using SDS-PAGE. (**C**) Cap protein analysis using Western blotting.

**Figure 7 vaccines-13-00801-f007:**
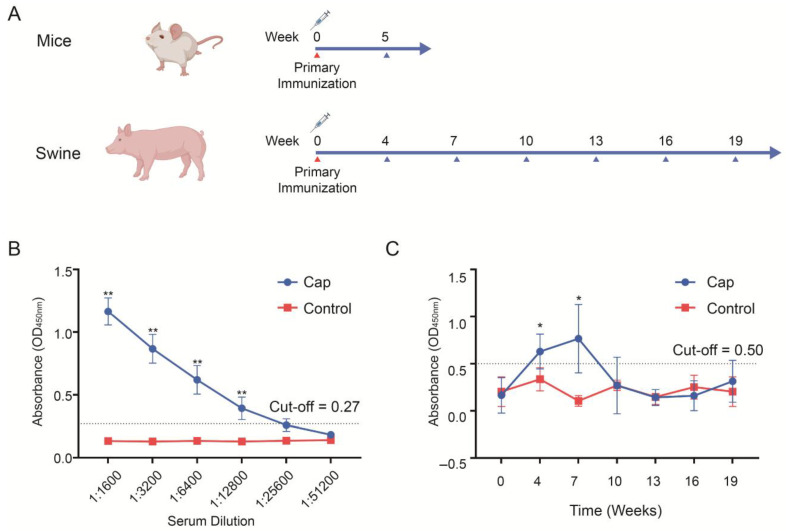
Serum antibody content in mice and swine after immunization. (**A**) Animal immunization and serum collection timeline. (**B**) Post-immunization IgG antibody in mice. (**C**) Post-immunization IgG antibody dynamics in swine. Asterisks indicate statistically significant differences (* *p* < 0.05, ** *p* < 0.01) by Student’s *t*-test.

## Data Availability

The datasets supporting the conclusions in this article are included within the article.

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
