# Peer review of "High-Yield Production of PCV2 Cap Protein: Baculovirus Vector Construction and Cultivation Process Optimization"

_vaccines, 2025, doi:10.3390/vaccines13080801_

Round 1

Reviewer 1 Report

Comments and Suggestions for Authors

This manuscript compares the productivity of PCV2 Cap protein expressed in two different baculovirus expression systems: the traditional Bac-to-Bac system and the flashBAC system. The results demonstrate that the flashBAC system achieves higher Cap protein expression levels when using the same insect cell line. Furthermore, the expression level of the target protein in High Five cells was consistently higher than in Sf9 cells across both expression systems, which aligns with findings from previous studies. Notably, the flashBAC system yielded the highest protein production when utilizing four copies of the Cap protein expression cassettes in High Five cells. These findings provide valuable insights for researchers using baculovirus expression systems for heterologous protein production and offer practical strategies for optimizing protein expression.
However, the manuscript requires several improvements prior to publication:
1. The background section should provide more detailed comparisons of the two baculovirus expression systems, particularly regarding genomic differences (e.g., specific gene deletions in flashBAC that may enhance expression levels).
2. Transmission electron micrographs (TEM) of the VLPs formed by PCV2 Cap protein are essential for structural characterization and should be included.
3. The marginal 12% increase in expression with four gene copies versus fewer copies warrants further investigation. Testing two copies might be more optimal due to potential promoter competition effects.
4. A supplementary neutralization assay evaluating PCV2-specific antibody titers induced by the recombinant Cap protein would strengthen the immunological relevance of the study.
5. The Discussion section requires substantial revision. The authors should comprehensively explain the observed expression differences by addressing: (a) how flashBAC's genomic modifications (e.g., deletion of proteolytic genes) enhance expression, and (b) how potential mutations accumulated during Bac-to-Bac system propagation in bacterial hosts might reduce expression efficiency.

Author Response

Comments 1: The background section should provide more detailed comparisons of the two baculovirus expression systems, particularly regarding genomic differences (e.g., specific gene deletions in flashBAC that may enhance expression levels).

Response 1: Add background introduction on flashBAC to the third paragraph of the Introduction.

The flashBAC system offers several modified baculoviral backbone vectors. The original flashBAC vector has the chiA (chitinase) gene deleted. The flashBAC Gold vector removes both the chiA and v-cath (cathepsin) genes. The flashBAC Ultra vector further deletes additional viral genes (such as p10, p74, and p26) beyond chiA and v-cath. These modifications reduce recombinant protein degradation and enable prolonged protein expression.

Comments 2: Transmission electron micrographs (TEM) of the VLPs formed by PCV2 Cap protein are essential for structural characterization and should be included.

Response 2: Transmission Electron Microscopy (TEM) is indeed the most direct method to demonstrate the formation of Virus-Like Particles (VLPs) by the Cap protein. The absence of TEM images showing Cap forming VLPs in the article is a significant shortcoming. The PCV2 Cap protein subunit vaccine technology used in this study was transformed from Zhaofenghua Bio-Tech Co., Ltd (兆丰华生物技术有限公司). Importantly, Zhaofenghua Bio-Tech Co., Ltd. has already documented TEM images of the Cap protein forming VLPs in their patent (visible in the attached patent documentation). The expression sequence of the Cap protein used in this experiment is identical to that of the production seed strain used by Zhaofenghua. Therefore, TEM imaging of the Cap protein was not repeated in this study. Cap protein produced using this production strain has already obtained the New Veterinary Drug Registration Certificate from China and has been granted a production approval number (see attached supporting documentation). Yunnan Biological Pharmaceutical Co., Ltd. (云南生物制药有限公司) is a subsidiary of Zhejiang Hisun Animal Health Products Co., Ltd.

Comments 3: The marginal 12% increase in expression with four gene copies versus fewer copies warrants further investigation. Testing two copies might be more optimal due to potential promoter competition effects.

Response 3: The impact of gene copy number on gene expression levels was not studied in detail in this experiment. The experiment only included single-copy and four-copy groups, lacking a double-copy group. In the future, when optimizing Cap expression or using the flashBAC system to express other antigen proteins, we will set up single-copy, double-copy, and four-copy groups to investigate the marginal effect of gene copy number on protein expression levels.

Comments 4: A supplementary neutralization assay evaluating PCV2-specific antibody titers induced by the recombinant Cap protein would strengthen the immunological relevance of the study.

Response 4: The article lacks data on neutralizing antibodies, which is indeed regrettable. However, we conducted a challenge experiment after vaccine immunization. The challenge method was as follows: 15 healthy susceptible pigs aged 3–4 weeks were divided into 3 groups, with 5 pigs in each group. Group 1 was the immunized group, with each pig receiving a 2.0 ml intramuscular injection of the vaccine in the neck. Group 2 was the challenge control group (non-immunized), and Group 3 was the blank control group (non-immunized, non-challenged). All pigs were kept in isolation. On day 28 post-immunization, all pigs were weighed. Groups 1 and 2 were then challenged with the PCV2DBN-SX07 strain (containing 106 TCID50/ml) via intramuscular injection of 4.0 ml and kept in isolation. After the challenge, the pigs were observed continuously for 28 days. On day 28 post-challenge, they were weighed again, and serum samples were collected. Daily weight gain was calculated, and qPCR was used to detect the presence of the virus in the serum.

Since the challenge experiment only briefly examined the daily weight gain of the pigs post-challenge and the viral load in the serum on day 28 post-challenge, without histopathological examination of various tissues, and considering that this study primarily focused on the construction of the baculovirus and the improvement of protein expression levels, these data were not included in the article. The figure below shows the challenge data.

Comments 5: The Discussion section requires substantial revision. The authors should comprehensively explain the observed expression differences by addressing: (a) how flashBAC's genomic modifications (e.g., deletion of proteolytic genes) enhance expression, and (b) how potential mutations accumulated during Bac-to-Bac system propagation in bacterial hosts might reduce expression efficiency.Response 5: (a)The flashBAC system offers several modified baculoviral backbone vectors. The original flashBAC vector has the chiA (chitinase) gene deleted. The flashBAC Gold vector removes both the chiA and v-cath (cathepsin) genes. The flashBAC Ultra vector further deletes additional viral genes (such as p10, p74, and p26) beyond chiA and v-cath. These modifications reduce recombinant protein degradation and enable prolonged protein expression. Our results demonstrate that the baculovirus constructed using flashBAC was more suitable for Cap protein expression than that constructed using the Bac-to-Bac system, regardless of whether Sf9 or High Five cells were employed. The cause of this result may be related to the use of the flashBAC Gold vector in this experiment. Study has shown that the presence of the chiA and v-cath genes can promote cell lysis, leading to a shortened protein expression time and a significant decrease in protein expression levels [28]. Deleting the chiA and v-cath genes can significantly increase the expression of the recombinant protein acetyl esterase enzyme [29].

This section has been added to the discussion.

(b) How potential mutations accumulated during passage in the bacterial host might reduce expression efficiency in the Bac-to-Bac system.
Potential mutations accumulated during passage of the bacmid in the host bacteria were not observed in the Bac-to-Bac system. However, studies have shown that baculovirus expression decreases with increasing passage number. The reason for this phenomenon is the inherent instability of the baculovirus genome. As the number of passages increases, mutations occur, leading to the generation of deletion mutants ("empty viruses") that lack the expression cassette for the target protein. Since these empty viruses do not bear the burden of target protein expression, they replicate faster than viruses expressing the target protein. Consequently, after multiple passages, the expression level of the target protein decreases. Therefore, in production, only virus stock within five passages is used.

This section has already been discussed in the text.

Reviewer 2 Report

Comments and Suggestions for Authors

The manuscript entitled “High-Yield Production of PCV2 Cap Protein: Baculovirus Vector Construction and Cultivation Process Optimization” compares two commonly used baculovirus construction systems for expressing the Cap protein in various insect cell lines. The study is well-structured and addresses an important topic. However, the following concerns should be addressed to improve the quality and clarity of the manuscript:

  1. Please convert the abstract into a structured format of approximately 250 words, including the following headings: Background/Objectives, Methods, Results, and Conclusions.
  2. Please provide a high-resolution version of Figure 1, as the current image is not very clear.
  3. Please revise the title of Section 2to “Materials and Methods.”
  4. Kindly include relevant referencesin the Materials and Methods section to support your experimental procedures.
  5. In Section 2.7: Animal Immunity and Efficacy Evaluation, please clearly describe all the mice groupsand the specific treatments they received.
  6. The adjuvant usedin the immunization experiments should be clearly stated in the manuscript.
  7. Please adjust the title of Section 3to “Results.”
  8. In Line 228, do you mean “Figure 2A”? Please revise accordingly.
  9. In Figure 2G, why is the marker faintwhile the sample bands appear strong? Please provide an explanation or improve the image quality.
  10. In Figure 3, both the marker and the bands appear faint—please clarify the reason or enhance the image contrast.
  11. For Figure 3, please add a scale barto the immunofluorescence assay (IFA) images to aid interpretation.
  12. Please provide a high-resolution version of Figure 4, as it is currently unclear.
  13. Please clarify why a challenge experimentwas not conducted after immunization with the purified Cap protein. Including such data or explaining the rationale for its exclusion would strengthen the manuscript.

Author Response

Comments 1: Please convert the abstract into a structured format of approximately 250 words, including the following headings: Background/Objectives, Methods, Results, and Conclusions.

Response 1: The abstract has been rewritten as requested into the sections Background/Objectives, Methods, Results, and Conclusions. Please find the details below.

Background/Objectives

Porcine circovirus disease (PCVD), caused by PCV2 infection, is a globally prevalent and economically significant immunosuppressive disease in swine, leading to syndromes such as post-weaning multisystemic wasting syndrome (PMWS) and porcine dermatitis and nephropathy syndrome (PDNS). The Cap protein of PCV2, a major protective antigen, self-assembles into virus-like particles (VLPs) in baculovirus expression systems. However, few studies have compared Cap protein expression across different baculovirus construction systems. This study aimed to evaluate and compare Cap protein expression in two widely used baculovirus construction systems across insect cell lines to optimize yield for cost-effective vaccine production.

Methods

The Cap protein was expressed using two commercial baculovirus construction systems—flashBAC and Bac-to-Bac—in multiple insect cell lines. Expression levels were quantified, and constructs containing four copies of the Cap protein gene were generated and evaluated. High Five cells were cultivated at 27°C, and yields were measured up to 5 days post-infection (dpi). Purified Cap protein was administered to mice and swine to assess immunogenicity via specific antibody production.

Results

The flashBAC system outperformed Bac-to-Bac, yielding higher Cap protein levels. The highest yield (432 μg/mL at 5 dpi) was achieved using the four-copy Cap construct in High Five cells at 27°C. Animal trials confirmed that purified Cap protein induced robust specific antibodies in both mice and swine, validating its immunogenic potential.

Conclusions

The flashBAC system significantly enhances Cap protein expression in High Five cells, especially with multi-copy constructs. This optimization reduces PCV2 vaccine production costs and enhances manufacturing efficiency.

Comments 2: Please provide a high-resolution version of Figure 1, as the current image is not very clear.

Response 2: Parts of the image elements in Figure 1 are not vector graphics and become blurry when magnified significantly. We have enhanced the clarity of the non-high-definition elements. The figure was then composited in Adobe Illustrator and saved as a PDF to maximize image clarity (see updated Figure 1). We sincerely regret any lack of clarity in the original image.

Comments 3: Please revise the title of Section 2to “Materials and Methods.”

Response 3:The title has been changed from "Material and Method" to "Materials and Methods".

Comments 4: Kindly include relevant referencesin the Materials and Methods section to support your experimental procedures.

Response 4: References have been added to support the following procedures in the Materials and Methods section: Bacmid construction.

Comments 5: In Section 2.7: Animal Immunity and Efficacy Evaluation, please clearly describe all the mice groupsand the specific treatments they received.

Response 5: Six-week-old BALB/c mice (Sipeifu, Beijing, China) were randomly divided into two groups (n = 5 per group). The vaccine group was immunized with 50 μg of Cap protein mixed with the Oil-in-Water emulsion adjuvant SDA 15A (Merchinade, SDA15AVG, Beijing, China) at a 4:1 ratio (protein:adjuvant, vol/vol). The control group received phosphate-buffered saline (PBS) mixed with the same adjuvant at a 4:1 ratio (PBS:adjuvant, vol/vol), administered at the same volume as the vaccine group.

Comments 6: The adjuvant usedin the immunization experiments should be clearly stated in the manuscript.

Response 6: The adjuvant used in the immunization experiments was Oil-in-Water emulsion adjuvant SDA 15A (Merchinade, SDA15AVG, Beijing, China).

Comments 7: Please adjust the title of Section 3to “Results.”

Response 7:The title has been changed from "Result" to "Results."

Comments 8: In Line 228, do you mean “Figure 2A”? Please revise accordingly.

Response 8: Line 228 has been modified to "Figure 2A."

Comments 9: In Figure 2G, why is the marker faintwhile the sample bands appear strong? Please provide an explanation or improve the image quality.

Response 9:The high concentration of the target gene fragment caused the machine's automatic exposure to use a shorter exposure time, resulting in blurry marker bands. The brightness of Figure 2G has been increased to make the marker bands clearer.

Comments 10: In Figure 3, both the marker and the bands appear faint—please clarify the reason or enhance the image contrast.

Response 10: The machine's automatic exposure used a short exposure time, causing the marker bands to appear blurry. The brightness of Figure 3 has been increased to make both the marker and sample bands clearer.

Comments 11: For Figure 3, please add a scale barto the immunofluorescence assay (IFA) images to aid interpretation.

Response 11: In Figure 3, since all IFA images are scaled consistently, a scale bar has been added only to the first image to help interpret the post-infection cell state.

Comments 12: Please provide a high-resolution version of Figure 4, as it is currently unclear.

Response 12: Redraw Figure 4 and compose the figure in Adobe Illustrator before saving it as a PDF to ensure the resolution meets the standards of the reviewers and the journal.

Comments 13: Please clarify why a challenge experimentwas not conducted after immunization with the purified Cap protein. Including such data or explaining the rationale for its exclusion would strengthen the manuscript.

Response 13: The challenge experiment after vaccine immunization. The challenge method was as follows: 15 healthy susceptible pigs aged 3–4 weeks were divided into 3 groups, with 5 pigs in each group. Group 1 was the immunized group, with each pig receiving a 2.0 ml intramuscular injection of the vaccine in the neck. Group 2 was the challenge control group (non-immunized), and Group 3 was the blank control group (non-immunized, non-challenged). All pigs were kept in isolation. On day 28 post-immunization, all pigs were weighed. Groups 1 and 2 were then challenged with the PCV2DBN-SX07 strain (containing 106 TCID50/ml) via intramuscular injection of 4.0 ml and kept in isolation. After the challenge, the pigs were observed continuously for 28 days. On day 28 post-challenge, they were weighed again, and serum samples were collected. Daily weight gain was calculated, and qPCR was used to detect the presence of the virus in the serum.

Since the challenge experiment only briefly examined the daily weight gain of the pigs post-challenge and the viral load in the serum on day 28 post-challenge, without histopathological examination of various tissues, and considering that this study primarily focused on the construction of the baculovirus and the improvement of protein expression levels, these data were not included in the article. The figure below shows the challenge data.

Reviewer 3 Report

Comments and Suggestions for Authors

PCV2 has been associated with multiple clinical presentations, collectively referred to as PCV-associated disease (PCVAD/PCVD). The most common clinical presentations associated with PCV2 infection are wasting or PCV2-systemic disease (PMWS/PCV2-SD), reproductive disease (PCV2-RD), and porcine dermatitis and nephropathy syndrome (PDNS). Although commercial vaccines are available, PCVAD/PCVD is still considered one of the most economically devastating diseases in the global swine industry. Cap, as the protective antigen of porcine circovirus, has been expressed in various proteins expression systems, but most commercial vaccines are based on recombinant proteins (virus-like particles) expressed in the insect baculovirus system. In this study, the authors compared two different baculovirus construction systems, Bac-to-Bac and flash-BAC, to generate recombinant baculoviruses expressing the Cap protein. Additionally, the flashBAC system was employed to construct baculoviruses with varying copy numbers of the Cap protein.
Although in my opinion the study is interesting, the text of the manuscript should be corrected for publication. 
1. The text in the Materials and Methods section should be in the past tense. Therefore, the text of subsections 2.3 and 2.7 should be corrected. Authors should carefully check the text of the entire manuscript, since such inaccuracies are common.
2. How many independent in vitro experiments were conducted? Since there is no information about this and the results of statistical data processing, the obtained results may be due to errors or mistakes. The authors should add this information.
3. Is a decrease in the titer of antibodies against the PСV2 virus in pigs at 10 weeks after immunization normal?

Author Response

Comments 1: The text in the Materials and Methods section should be in the past tense. Therefore, the text of subsections 2.3 and 2.7 should be corrected. Authors should carefully check the text of the entire manuscript, since such inaccuracies are common.

Response 1: We have revised the writing errors in sections 2.3 and 2.7 as per your request. Additionally, the entire manuscript has undergone professional language polishing by the journal's editing team to enhance readability. Attached is the proof of language refinement.

Comments 2: How many independent in vitro experiments were conducted? Since there is no information about this and the results of statistical data processing, the obtained results may be due to errors or mistakes. The authors should add this information.

Response 2: To enhance the precision of the immunological data, we compared vaccines from multiple manufacturers within China with those from outside the region. The results were further validated against data provided by the product transfer unit, Zhao Fengnian Technology Co., Ltd. The Cap protein sequence used in this experiment matched the expression sequence of the production strain's Cap protein. Cap protein produced using this production strain has already obtained the New Veterinary Drug Registration Certificate from China and has been granted a production approval number (see attached supporting documentation). Yunnan Biological Pharmaceutical Co., Ltd. (云南生物制药有限公司) is a subsidiary of Zhejiang Hisun Animal Health Products Co., Ltd.

Comments 3: Is a decrease in the titer of antibodies against the PСV2 virus in pigs at 10 weeks after immunization normal?

Response 3: To enhance the accuracy of the immune data, we simultaneously compared vaccines from multiple manufacturers within China and those outside the Chinese region. The results are as follows. When immunizing pigs, we administered antibodies from multiple manufacturers simultaneously. The results showed similar trends across different manufacturers, with antibody titers declining after 10 weeks. This trend is consistent with the data from other companies and the original transfer company, Zhao Fengnian Technology Co., Ltd., where a decline was also observed after 10 weeks.

Round 2

Reviewer 2 Report

Comments and Suggestions for Authors

The quality of the current manuscript has been improved greatly. I now agree for further process. Congratulations!

Author Response

No revisions were made as the reviewer had no feedback.

Reviewer 3 Report

Comments and Suggestions for Authors

The changes made have improved the quality of the manuscript. The manuscript can be published and will be useful and interesting to other researchers.

Author Response

(The authors gave the same response as above.)
